# Mitochondria-Associated Membranes as Key Regulators in Cellular Homeostasis and the Potential Impact of Exercise on Insulin Resistance

**DOI:** 10.3390/ijms25063196

**Published:** 2024-03-11

**Authors:** Xi Li, Yangjun Yang, Xiaoyu Shi, Zhe Zhang, Shuzhe Ding

**Affiliations:** 1Key Laboratory of Adolescent Health Assessment and Exercise Intervention of Ministry of Education, East China Normal University, Shanghai 200241, China; LiXi989508060506@163.com (X.L.); yjyang98@163.com (Y.Y.); 2College of Physical Education & Health, East China Normal University, Shanghai 200241, China; xyshi1221@126.com

**Keywords:** mitochondria-associated membranes, insulin resistance, exercise, calcium homeostasis, lipid homeostasis, mitochondrial quality control, endoplasmic reticulum stress

## Abstract

The communication between mitochondria and the endoplasmic reticulum (ER) is facilitated by a dynamic membrane structure formed by protein complexes known as mitochondria-associated membranes (MAMs). The structural and functional integrity of MAMs is crucial for insulin signal transduction, relying heavily on their regulation of intracellular calcium homeostasis, lipid homeostasis, mitochondrial quality control, and endoplasmic reticulum stress (ERS). This article reviews recent research findings, suggesting that exercise may promote the remodeling of MAMs structure and function by modulating the expression of molecules associated with their structure and function. This, in turn, restores cellular homeostasis and ultimately contributes to the amelioration of insulin resistance (IR). These insights provide additional possibilities for the study and treatment of insulin resistance-related metabolic disorders such as obesity, diabetes, fatty liver, and atherosclerosis.

## 1. Introduction

With the improvement in people’s quality of life and changes in lifestyle, the incidence of metabolic diseases such as obesity, diabetes, fatty liver, and atherosclerosis has rapidly increased. Investigating their etiology reveals a common pathological mechanism—insulin resistance (IR). IR is characterized by decreased sensitivity of body tissues and cells to insulin, leading to compensatory insulin secretion by pancreatic β-cells and ultimately resulting in hyperinsulinemia. The blockade of the insulin signaling pathway is a major cause of IR, closely associated with disruptions in lipid homeostasis, calcium homeostasis, mitochondrial dysfunction, and endoplasmic reticulum stress (ERS) [1,2].

Recently, research has identified a liquid ordered structure with unique biophysical characteristics formed between mitochondria and the endoplasmic reticulum (ER), known as mitochondria-associated membranes (MAMs). MAMs are closely associated with the regulation of cellular homeostasis processes, including calcium homeostasis, lipid homeostasis, cell survival, inflammation, ERS, and mitochondrial quality control [2,3]. Key proteins in the insulin signaling pathway, such as protein kinase B (PKB/AKT), mammalian target of rapamycin complex 2 (mTORC2) [4], and phosphatase and tensin homolog (PTEN) [5], not only localize to MAMs under specific physiological conditions but also interact with resident MAMs proteins. This suggests that MAMs may play a crucial role in regulating insulin signal transduction.

Exercise, as a green and economic therapeutic approach, has been shown to have a definite beneficial effect on IR, although the specific molecular mechanisms remain unclear. However, studies indicate that exercise can regulate the expression of various cellular homeostasis regulatory factors associated with the structure and function of MAMs. It is speculated that there may be a close connection among MAMs, exercise, and IR. Based on this, this paper will review the relationship between MAM-related cellular homeostasis regulation and IR, as well as the potential mechanisms through which MAMs mediate exercise intervention in IR, aiming to provide new targets for the treatment of IR-related metabolic diseases.

## 2. Overview of Mitochondria-Associated Membranes

The formation of intracellular membranes plays a crucial role in the evolution of species, providing a prerequisite guarantee for membrane-bound organelles to fulfill their specific functions. As a complex entity, the cell relies on the synergistic interactions among various organelles to achieve its intricate functions. Effective cellular communication is a fundamental prerequisite for the collaboration among organelles, and this relies on the involvement of organelle membranes. Cellular communication encompasses various mechanisms, one of which is achieved through membrane contact sites (MCSs). The most classic example of this is the membrane contact between the endoplasmic reticulum (ER) and mitochondria, initially referred to as “X components” and later named mitochondria-associated membranes (MAMs) [6]. Recent studies suggest that the minimum distance between MAMs interfaces may reach 10–25 nm [7], allowing for direct contact between ER proteins and the proteins and lipids on the outer membrane of mitochondria, providing various possibilities for inter-organelle information exchange.

MAMs are widely distributed in different organisms. In yeast cells, the corresponding structure of MAMs is known as ER–mitochondria encounter structure (ERMES). It is composed of four proteins, Mdm12, Mdm34, Mdm10, and Mmm1, participating in the mediation of lipid transfer facilitated by soluble lipid carrier proteins such as ceramide transfer protein (CERT) and oxysterol binding protein (OSBP) [8]. In contrast, mammalian MAMs have a more complex connecting structure formed by single or multiple tethering proteins (As shown in Figure 1). These tethering proteins not only participate in the formation of MAMs but are also closely associated with their functions. Tethering is just one condition for maintaining the structural and functional integrity of MAMs; many other proteins participate in the formation and influence the function of MAMs in a tether-independent manner, such as Phosphofurin acidic cluster sorting protein 2 (PACS-2), PDZ domain containing 8 (PDZD8), sigma-1 receptor (Sig-1R), FUN14 domain-containing 1 (FUNDC1), and others. Under normal circumstances, the structure and function of MAMs are maintained in a dynamic equilibrium. However, under specific physiological conditions, the molecules mentioned above, through an unknown mechanism, promote the remodeling of the MAMs network. This, in turn, regulates various life processes, such as intracellular calcium homeostasis, lipid metabolism and transport, autophagy, apoptosis, mitochondrial quality control, and endoplasmic reticulum stress (ERS). Subsequently, these processes further influence organismal metabolism and the occurrence and development of related diseases.

## 3. Mitochondria-Associated Membranes Intervene in Insulin Resistance through Cellular Homeostasis

Mitochondria-associated Membranes (MAMs), as a type of lipid raft structure, possess various binding domains for proteins involved in signal transduction, including transport proteins, kinases, ion channels, or phosphatases. This provides MAMs with the potential to regulate numerous physiological and pathological processes within the cell, such as calcium homeostasis, lipid homeostasis, mitochondrial quality control, and ERS. Recent studies indicate that the above-mentioned processes mediated by MAMs appear to be associated with insulin resistance (IR).

### 3.1. Mitochondria-Associated Membranes Intervene in Insulin Resistance through Calcium Homeostasis

Calcium ion (Ca^2+^) homeostasis refers to the intracellular low-calcium state maintained through plasma membrane calcium turnover and calcium reservoir regulation under normal physiological conditions. Numerous Ca^2+^ transport channels, such as mitochondrial calcium uniporter (MCU), ryanodine receptors (RyRs), and the inositol 1,4,5-triphosphate receptors-glucose-regulated protein 75-voltage-dependent anion channel 1 (IP_3_Rs-Grp75-VDAC1) complex, are distributed on mitochondria-associated membranes (MAMs) [9,10,11]. Additionally, proteins involved in calcium regulation, such as sigma-1 receptor (Sigma-1R), Mitofusin2 (Mfn2), FUN14 domain-containing 1 (FUNDC1), and calnexin (CNX), are also enriched in MAMs. Therefore, MAMs may serve as crucial components in regulating intracellular calcium reservoirs [12].

Studies reveal that the regulation of intracellular Ca^2+^ homeostasis by MAMs is a key factor influencing insulin resistance (IR). MAMs enrichment in hepatocytes of obese mice and oocytes leads to mitochondrial Ca^2+^ overload [13], resulting in excessive mitochondrial fission and IR [14]. Knocking out Phosphofurin acidic cluster sorting protein 2 (PACS-2), a protein associated with MAMs formation, reduces MAMs formation and decreases mitochondrial Ca^2+^ levels [13]. However, structural disruption of MAMs leads to excessive Ca^2+^ release into the cytoplasm, causing cytoplasmic Ca^2+^ waves and inducing gluconeogenesis, followed by IR. This indicates that excessive formation or structural disruption of MAMs may lead to IR through different pathways. Additionally, MAMs-related Ca^2+^ signal transduction is a critical event in insulin-dependent glucose uptake. Research suggests that the G protein/IP_3_/IP_3_R pathway regulates the fusion of glucose transporter 4 (GLUT4) with the plasma membrane and that Ca^2+^ chelators inhibit drug-induced GLUT4–plasma membrane fusion and glucose uptake [15].

In conclusion, MAMs maintain appropriate Ca^2+^ flux among the endoplasmic reticulum (ER), cytoplasm, and mitochondria under normal physiological conditions. When the body experiences nutritional or energy overload, MAMs undergo network remodeling, leading to MAMs enrichment causing mitochondrial Ca^2+^ overload, promoting excessive mitochondrial fission, and inducing IR. Structural disruption of MAMs, on the one hand, blocks Ca^2+^ signal transduction and GLUT4 translocation, inhibiting muscle glucose uptake; on the other hand, it leads to insufficient mitochondrial Ca^2+^ absorption and cytoplasmic Ca^2+^ overload, thereby inducing IR caused by gluconeogenesis.

### 3.2. Mitochondria-Associated Membranes Modulate Insulin Resistance through Lipid Homeostasis

Aberrant lipid metabolism is closely associated with insulin resistance (IR). Studies have demonstrated that inhibiting elevated levels of ceramides in obese rats can improve hypothalamic insulin sensitivity and prevent central IR [16]. Impairment in the insulin signaling pathway may further lead to elevated ceramide levels and ceramide-induced atypical protein kinase C (PKC) activation, exacerbating IR [17]. Mitochondria-associated membranes (MAMs) are lipid raft-like domains enriched with cholesterol and sphingolipids within cells. Numerous enzymes and proteins involved in lipid synthesis, transport, and metabolism are enriched in MAMs (Figure 2). It is suggested that MAMs are likely to be crucial for the regulation of intracellular lipid homeostasis.

Indeed, MAMs are not only implicated in the metabolism and transport of phospholipids and cholesterol [18] but also serve as critical pathways for the transport of ceramides from the endoplasmic reticulum (ER) to mitochondria. Even in yeast cells, the MAMs equivalent, known as ER–mitochondria encounter structure (ERMES), serves as a foundation for efficient lipid transport by soluble lipid carrier proteins like ceramide transfer protein (CERT) and oxysterol binding protein (OSBP) [8].

Recent research has unveiled a close interrelation among MAMs, lipid homeostasis, and IR. It has been reported that compromised MAMs integrity leads to reduced fatty acid oxidation and increased levels of fatty acyl-CoA and diacylglycerol (DAG), subsequently upregulating Ser/Thr kinase activity. This results in enhanced serine phosphorylation of Insulin Receptor Substrate 1 (IRS-1) and impedes tyrosine phosphorylation of IRS-1 by the insulin receptor, ultimately inhibiting insulin-induced glucose uptake [19]. Additionally, studies indicate that under normal conditions, ceramides can be transferred to mitochondria and converted into sphingosine-1-phosphate and hexadecenal. However, disruption in ER–mitochondrial coupling prevents ceramide transfer to mitochondria, leading to elevated cytosolic ceramide levels [20]. The positioning of ceramides near the plasma membrane is a critical factor inhibiting insulin signaling [21]. Thus, there is compelling evidence to suggest that MAMs may ensure appropriate insulin sensitivity by maintaining intracellular lipid homeostasis.

### 3.3. The Role and Mechanism of Mitochondrial Quality Control in Mitochondria-Associated Membranes Intervention in Insulin Resistance

Mitochondria serve as crucial sites for cellular metabolism, participating in the regulation of organismal nutrition and energy status. Mitochondrial dysfunction is a common pathogenic feature of insulin resistance (IR) [1], and the intracellular mitochondrial quality control system can jointly regulate mitochondrial function through physiological processes such as mitochondrial biogenesis, mitochondrial dynamic changes, and mitophagy. However, the specific connection between mitochondria and insulin sensitivity is not yet clear. Recent studies suggest that mitochondria-associated membranes (MAMs) may play a crucial role in this context.

#### 3.3.1. Mitochondrial Biogenesis, Mitochondria-Associated Membranes, and Insulin Resistance

Mitochondrial biogenesis is the process of producing new, well-functioning mitochondria within cells and is a key step for mitochondria to function normally. Peroxisome proliferator-activated receptor gamma coactivator 1-alpha (PGC-1α) is a key transcription factor for mitochondrial biogenesis primarily coordinating this process by promoting the transcription of nuclear genes and mitochondrial DNA (mtDNA) [22]. Research indicates that PGC-1α, located on chromosome 4p15.1-2, correlates with basal insulin levels in different populations [1]. Additionally, PGC-1α levels are reduced in the skeletal muscle of individuals with insulin resistance (IR) and type 2 diabetes mellitus (T2DM) [23]. Therefore, the downregulation of PGC-1α expression leading to decreased mitochondrial biogenesis could be one of the factors contributing to IR. Notably, PGC-1α expression increases with exercise, further promoting mitochondrial biogenesis [24], suggesting that exercise may intervene in IR through PGC-1α-mediated mitochondrial biogenesis. Interestingly, exposure to perfluorooctane sulfonic acid (PFOS) in mouse cardiomyocytes activates mammalian target of rapamycin complex 2 (mTORC2) by phosphorylating epidermal growth factor receptor (EGFR) (Tyr1086), weakening the tethered the inositol 1,4,5-triphosphate receptors-glucose-regulated protein 75-voltage-dependent anion channel (IP_3_R-Grp75-VDAC) interaction on mitochondria-associated membranes (MAMs), leading to intracellular fatty acid accumulation and subsequent reduction in PGC-1α expression, resulting in decreased mitochondrial biogenesis [25]. This suggests that the weakened interaction of MAMs tethering complex proteins is a potential key regulator causing the downregulation of PGC-1α, leading to reduced mitochondrial biogenesis and consequent IR.

#### 3.3.2. Mitochondrial Dynamics, Mitochondria-Associated Membranes, and Insulin Resistance

Mitochondrial dynamics involve the balance of fusion and fission processes determining the structure and distribution of mitochondria [26]. Various proteins, including mitochondrial fission protein 1 (Fis1), Mitochondrial Fission Factor (Mff), Opticatrophy 1 (OPA1), Mitofusin 1 (Mfn1), Mitofusin 2 (Mfn2), and Dynamin-Related Protein 1 (Drp1), play crucial roles in maintaining the balance of mitochondrial dynamics, and interestingly, all these molecules are detected in mitochondria-associated membranes (MAMs) [27]. This suggests a potential connection between MAMs and the regulation of mitochondrial dynamics.

In fact, researchers have identified the critical role of endoplasmic reticulum (ER) in mitochondrial fission [28] and fusion [29] in recent years. Research indicates that FUN14 domain-containing 1 (FUNDC1) expression is essential for MAMs formation, and the loss of FUNDC1 in cardiac muscle downregulates Fis1 expression, inhibiting mitochondrial fission, while overexpression of FUNDC1 leads to excessive mitochondrial fission [14]. MAMs can also regulate the expression of genes related to mitochondrial dynamics at the transcriptional level. The increased formation of MAMs leads to an increase in cytoplasmic Ca^2+^ concentration, thereby activating the calcium-sensitive transcription factor cAMP-response element binding protein (CREB). Activated CREB directly binds to the promoter of Fis1, promoting its transcription and thereby enhancing mitochondrial fission [30]. Moreover, the unique lipid environment of MAMs is believed to promote membrane curvature, facilitating membrane fission and fusion. Therefore, MAMs are crucial platforms for the regulation of mitochondrial dynamics.

Mfn2 is a classical functional tethering protein in MAMs, and its knockout or silencing leads to the dissociation of the endoplasmic reticulum (ER) and mitochondria [31]. Liver-specific knockout of Mfn2 causes excessive mitochondrial fission, reduces insulin signal transduction in muscle and liver tissues, and induces susceptibility to insulin resistance (IR) [32]. In myocardial cells, FUNDC1 binds to inositol 1,4,5-triphosphate receptor 2 (IP_3_R_2_) to form MAMs tethers, facilitating the transfer of ER Ca^2+^ to mitochondria and the cytoplasm [30]. Under normal physiological conditions, FUNDC1-knockout mice exhibit decreased Fis1 expression, leading to excessive mitochondrial fusion and heart dysfunction. However, under stimuli such as high fat and high sugar, Adenosine 5′-monophosphate (AMP)-activated protein kinase (AMPK) activity is inhibited, resulting in abnormal elevation in FUNDC1 and FUNDC1-mediated formation of MAMs, leading to abnormal accumulation of mitochondrial Ca^2+^, which further affects mitochondrial function through excessive mitochondrial fission. In this scenario, downregulation of the FUNDC1 gene can prevent excessive mitochondrial fission by inhibiting the excessive formation of MAMs and the associated elevation of mitochondrial Ca^2+^, thus preventing and improving diabetic cardiomyopathy [14]. Studies also suggest that exercise activates AMPK, which can improve lipid levels and IR [33]. This implies that exercise may alleviate high-sugar- and high-fat-induced IR by activating AMPK, downregulating FUNDC1 expression, inhibiting MAMs enrichment, preventing excessive mitochondrial fission.

In summary, under normal physiological conditions, MAMs maintain the dynamic balance of mitochondrial fission and fusion. When the body is in a state of energy or nutrient overload, the MAMs network may undergo remodeling. Enrichment of MAMs lead to mitochondrial Ca^2+^ overload, which triggers excessive mitochondrial fission, thus resulting in IR. However, some studies also suggest that changes in mitochondrial dynamics during nutrient overload are associated with mitochondrial depolarization. Therefore, the exact reasons for mitochondrial morphological changes under conditions of nutrient overload require further experimental verification.

#### 3.3.3. Mitophagy, Mitochondria-Associated Membranes, and Insulin Resistance

Mitochondria undergo frequent cycles of fusion and fission in a “kiss and run” pat-tern. The membrane potential of the daughter units generated during fission events typically undergoes distinct changes, determining their subsequent fate. Daughter units with increased membrane potential can be selectively fused by the mitochondrial network to repair damaged regions. On the other hand, daughter units with decreased membrane potential are unable to undergo fusion for repair and are more inclined to be eliminated through autophagy [34]. Mitophagy is a multi-step catabolic process that selectively targets damaged or dysfunctional mitochondria for lysosomal-dependent degradation, ensuring the stability of mitochondrial quality. Mitochondrial dysfunction has been implicated in the development of liver insulin resistance (IR) induced by the accumulation of fatty acids. Mitophagy selectively degrades damaged mitochondria to reverse mitochondrial dysfunction, inhibit hepatic fatty acid accumulation, and consequently improve liver IR. The mitophagy-related protein PTEN induced putative kinase 1 (PINK1) was downregulated in high-fat-fed mice, while overexpression of PINK1 enhanced glucose uptake and downregulated gluconeogenic enzyme levels [35]. This suggests that impaired mitophagy function is a significant contributing factor to high-fat-induced IR.

It is noteworthy that damaged mitochondria can also generate damage-associated molecular patterns (DAMPs), such as reactive oxygen species (ROS), to activate the Nucleotide-binding oligomerization domain, leucine-rich repeat and pyrin domain-containing 3 (NLRP3) inflammasome. Activated NLRP3 recruits the adaptor protein apoptosis-associated speck-like protein (ASC) to specifically localize to mitochondria-associated membranes (MAMs). Under pro-inflammatory stimuli, NLRP3 oligomerizes and exposes its effector domain to interact with ASC. ASC, in turn, recruits pro-caspase-1 to form an active NLRP3 inflammasome complex. Finally, activated caspase-1 cleaves IL-1β to form mature IL-1β [36]. Studies have shown that inflammasomes can directly or indirectly affect insulin signaling pathways, contributing to the development of IR and type 2 diabetes mellitus (T2DM) [37]. This suggests that MAMs may play an important role in the inflammatory insulin resistance caused by mitochondrial damage.

In yeast cells, the disruption of the ER–mitochondria encounter structure (ERMES) leads to reduced mitochondrial division and mitophagy. However, artificial restoration of the ERMES enables the recovery of mitophagy [38,39], indicating the involvement of the ERMES in regulating mitophagy. Indeed, in both mammals and yeast, MAMs/ERMES control the selective degradation of unused or damaged mitochondria and mark sites for mitochondrial division. FUN14 domain-containing 1 (FUNDC1), a key molecule in regulating MAMs formation, also serves as a mitophagy receptor. Under low-oxygen conditions, FUNDC1 binds to calnexin at MAMs, promoting mitophagy. As mitophagy progresses, FUNDC1 dissociates from calnexin, and Dynamin-Related Protein 1 (Drp1) binds to the exposed site, thereby being recruited to MAMs, leading to mitochondrial fission. Downregulation of FUNDC1, Drp1, or calnexin can inhibit mitochondrial division and mitophagy [40]. This allows mitochondrial fission and mitophagy to be integrated at the MAMs interface. In high-fat-fed mice, specific deletion of FUNDC1 in adipocytes impaired mitophagy, exacerbating obesity and IR [41]. Additionally, MAMs formation significantly increased in the oocytes of obese mice subjected to a high-fat diet [13].

In summary, under normal conditions, MAMs likely maintain the necessary level of mitophagy. When the organism is exposed to high-fat stimuli, the remodeling of the MAMs network increases the number of MAMs, enhancing mitophagy to eliminate a large number of functionally impaired mitochondria induced by high-fat stimuli. Despite the compensatory increases in MAMs under high-fat conditions promoting mitophagy, this does not completely compensate for the mitochondrial dysfunction caused by high-fat intake, resulting in the persistence of IR.

#### 3.3.4. Endoplasmic Reticulum Stress, Mitochondria-Associated Membranes, and Insulin Resistance

The endoplasmic reticulum (ER) is a crucial cellular organelle involved in maintaining calcium homeostasis, protein synthesis, post-translational modification, and transport. During insulin resistance (IR), the dissociation of glucose-regulated protein 78 (GRP78) from three ER membrane transport proteins—Protein kinase R-like endoplasmic reticulum kinase (PERK), Inositol-requiring enzyme-1α (IRE-1α), and Activating Transcription Factor 6 (ATF6)—occurs, leading to the unfolded protein response (UPR). The UPR can alleviate endoplasmic reticulum stress (ERS), but when ERS exceeds the reparative capacity of UPR, cells may undergo apoptosis [42].

ERS has been reported as a significant factor contributing to IR [43]. When ERS occurs, it activates the serine kinase c-Jun N-terminal kinase (JNK) and nuclear factor kappa B (NF-κB) signaling pathways, disrupting normal insulin signal transduction and causing IR [44]. Recently, studies have found that mitochondria-associated membranes (MAMs) appear to play a crucial role in the ERS promoting IR. Disruption of MAMs structure and functional integrity in liver cells exacerbates ERS, leading to IR [45]. Knocking out MAMs tethering protein Mitofusin 2 (Mfn2) results in the generation of reactive oxygen species (ROS) and ERS in the liver and muscles. The interaction between the two activates JNK, phosphorylating Insulin Receptor Substrate (IRS) proteins and inhibiting insulin signal transduction [32]. Overexpressing MAMs tethering proteins glucose-regulated protein 75 (GRP75) or Mfn2 in muscle cells suppresses palmitate-induced ERS, thereby improving IR [46]. Furthermore, the dynamics and regulation of MAMs contribute to the interaction between ERS and mitochondrial dysfunction in decreased insulin responsiveness [47]. This aligns with recent research indicating enhanced ER–mitochondria coupling in the early stages of ERS [48], but the uncoupling of the ER and mitochondria interrupts calcium transfer [31], leading to subsequent ERS [49].

Collectively, the evidence suggests that maintaining a dynamic balance of MAMs within a certain range is crucial to preventing ERS and maintaining normal insulin sensitivity under normal physiological conditions. Additionally, compensatory increases in MAMs can improve ERS-induced IR when the organism experiences ERS under stress.

## 4. Role and Mechanism of Mitochondria-Associated Membranes-Mediated Cellular Homeostasis in Exercise Intervention in Insulin Resistance

Exercise serves as a green and economically sustainable approach to intervening in insulin resistance (IR). Studies indicate that various forms of physical activity can enhance insulin sensitivity to a certain extent [50]. However, the precise mechanisms underlying exercise interference with IR remain unclear. Presently, research on exercise intervention in IR primarily focuses on functional changes within individual cellular organelles, with limited attention being given to the interactions among organelles. In recent years, scholars have started to investigate the role of inter-organelle interactions in exercise-regulated IR, with a significant emphasis on mitochondria-associated membranes (MAMs). Studies suggest that the beneficial effects of exercise on IR may be attributed to its regulation of MAMs-associated calcium ion (Ca^2+^) signaling, lipid homeostasis, mitochondrial quality control, and endoplasmic reticulum stress (ERS).

### 4.1. Mitochondria-Associated Membranes-Associated Calcium Ion Signaling Mediates Exercise Intervention in Insulin Resistance

Intracellular mitochondrial or cytoplasmic calcium ion (Ca^2+^) overload is a crucial factor contributing to insulin resistance (IR). Mitochondria-associated membranes (MAMs), acting as “assembly points” for Ca^2+^ channels, finely regulate intracellular Ca^2+^ transport and Ca^2+^ signal transduction. Disruption of the MAMs structure leads to excessive Ca^2+^ release into the cytoplasm, generating cytoplasmic Ca^2+^ waves, thereby inducing gluconeogenesis and causing IR [2]. Muscle contraction during exercise is a concrete manifestation of the transmission of electrical signals from motor neurons to the muscle cell membrane, the coupling of electrical signals to mechanical signals, and the sliding of thick and thin myofilaments. Current research suggests that the activation of Ca^2+^ channels on MAMs may be closely related to changes in muscle electrical activity. When skeletal muscle cell membranes depolarize, MAMs-associated Ca^2+^ channels [inositol 1,4,5-triphosphate receptor (IP_3_R)/ryanodine receptor 1 (RyR1)] are activated, leading to increased Ca^2+^ absorption by mitochondria [51]. Neuronal action potentials can stimulate the expression of MAMs-associated Ca^2+^ channel proteins, IP_3_Rs and RyRs, promoting endoplasmic reticulum (ER) Ca^2+^ release, subsequently facilitating mitochondrial Ca^2+^ uptake induced by another Ca^2+^ channel, mitochondrial calcium uniporter (MCU), on MAMs [52]. Interestingly, knockout of MCU was shown to impair mouse motor ability and decrease mitochondrial Ca^2+^ uptake [53]. Therefore, the use of exercise to interfere with IR through MAMs-related Ca^2+^ transport pathways may involve two potential mechanisms: (1) Exercise stimulation triggers a mechanism that repairs MAMs structure, leading to an increase in MAMs-associated Ca^2+^ channels within cells, promoting mitochondrial Ca^2+^ absorption, and inhibiting cytoplasmic Ca^2+^ deposition, thereby alleviating IR. This mechanism requires further exploration. (2) Exercise directly upregulates the expression of MAMs-associated Ca^2+^ channel proteins (IP_3_Rs/RyRs/MCU), promoting mitochondrial Ca^2+^ absorption, decreasing cytoplasmic Ca^2+^ deposition, and alleviating IR.

The Ca^2+^ signaling transduction associated with MAMs is a crucial event for insulin-dependent glucose uptake [15]. The G protein/IP_3_/IP_3_R pathway induces fusion of glucose transporter 4 (GLUT4) with the plasma membrane and subsequent glucose uptake in a Ca^2+^-dependent manner, increasing intracellular Ca^2+^ concentration [15]. Impairment of this Ca^2+^ transduction pathway inhibits GLUT4 translocation and subsequent glucose uptake. Exercise, by inducing Ca^2+^ signaling transduction, promotes GLUT4 translocation, and acute regulation of muscle glucose uptake also relies on the translocation and expression of GLUT4. Exercise training can effectively stimulate the expression of GLUT4 in skeletal muscle. This effect helps to improve insulin action to a certain extent [54]. Therefore, exercise likely activates the MAMs-related Ca^2+^ signaling pathway (G protein/IP_3_/IP_3_R), induces Ca^2+^ signal transduction, promotes GLUT4 translocation to the plasma membrane, and enhances muscle glucose uptake, thus alleviating IR.

### 4.2. Mitochondria-Associated Membranes-Associated Lipid Metabolism Mediates Exercise Intervention in Insulin Resistance

Mitochondria-associated membranes (MAMs) serve as lipid raft structures closely associated with lipid homeostasis, with many enzymes and proteins involved in lipid synthesis, transport, and metabolism being enriched in MAMs. The accumulation of lipids, especially diacylglycerol (DAG) and sphingolipids, in skeletal muscles is related to decreased insulin sensitivity in humans [55]. Studies indicate that disruption of MAMs integrity results in increased DAG levels, inhibiting insulin-induced glucose uptake. Moderate-and high-intensity aerobic exercise can reduce DAG levels [56]. Both acute or chronic aerobic and resistance exercise enhances insulin sensitivity [55], suggesting that exercise likely reduces DAG levels by repairing the MAMs structure, thereby alleviating insulin resistance (IR).

Research also suggests that the regulatory effect of exercise on triglycerides (TGs) in cells is also related to MAMs. Increased intramyocellular triglycerides (IMTGs) are closely associated with IR in obese and type 2 diabetes mellitus (T2DM) patients. In dietary obese mice and obese T2DM patients, disruption of MAMs was demonstrated to result in adverse effects, such as TG accumulation and IR [57]. However, chronic aerobic exercise training reduces IMTGs in obese individuals both with and without T2DM [56]. In addition, studies indicate that the enzyme acyl coenzyme A: diacylglycerol acyltransferase 2 (DGAT2), involved in triglyceride synthesis, is localized in MAMs. This suggests that exercise may alleviate IR by promoting the restoration of the MAMs structure in obesity or T2DM, inhibiting TG formation.

Moreover, ectopic deposition of ceramides also lead to IR. Structural disruption of MAMs leads to excessive accumulation of ceramide in the cytoplasm, resulting in IR [20]. We found that aerobic exercise training promotes the expression of MAMs-related proteins in the skeletal muscle of diabetic mice [58]. Moreover, research indicates that aerobic exercise or aerobic interval training reduces skeletal muscle ceramides in obese and T2DM populations [59,60]. This suggests that exercise may alleviate IR by repairing the MAMs structure, alleviating ectopic deposition of ceramides in obese and T2DM.

The above research results suggest the critical importance of MAMs in lipid metabolism in tissues such as skeletal muscle and liver under basal conditions. Under physiological or pathological conditions such as obesity or T2DM, disruption of the MAMs structure induces intracellular lipid accumulation or lipid degeneration. Exercise training can promote the repair of the MAMs structure and improve lipid metabolism, thus alleviating IR. Although numerous studies have demonstrated that exercise can upregulate the expression of MAMs-forming proteins such as Mitofusin2 (Mfn2), FUN14 domain-containing 1 (FUNDC1), and mammalian target of rapamycin complex 2 (mTORC2), the pathways through which exercise restores the MAMs structure are likely more complex and need further investigation, with direct evidence being still somewhat lacking.

### 4.3. Mitochondria-Associated Membranes-Associated Mitochondrial Quality Control Mediates Exercise Intervention Insulin Resistance

Disruption of the mitochondria-associated membranes (MAMs) structure in insulin resistance (IR) and type 2 diabetes mellitus (T2DM) patients results in downregulation of Peroxisome proliferator-activated receptor gamma coactivator 1-alpha (PGC-1α) expression, inhibiting mitochondrial biogenesis. Conversely, aerobic exercise can upregulate PGC-1α expression in IR patients, enhancing mitochondrial biogenesis and efficiency, thereby improving insulin sensitivity [61]. This suggests that aerobic exercise-induced improvement in insulin sensitivity in IR patients may be related to induced mitochondrial biogenesis through MAMs formation. Mammalian target of rapamycin complex 2 (mTORC2) likely serves as a key switch in exercise regulation of MAMs-associated mitochondrial biogenesis. Studies indicate that mouse myocardial cells exposed to perfluorooctanesulfonic acid (PFOS) activated mTORC2 through phosphorylation of epidermal growth factor receptor (EGFR) (Tyr1086), reducing the interaction of MAMs tethering inositol 1,4,5-triphosphate receptors-glucose-regulated protein 75-voltage-dependent anion channel (IP_3_R-Grp75-VDAC), leading to intracellular fatty acid accumulation and subsequently decreasing PGC-1α expression, inhibiting mitochondrial biogenesis [25]. However, under insulin stimulation, mTORC2 is located on MAMs and phosphorylates MAMs-residing proteins Phosphofurin acidic cluster sorting protein 2 (PACS-2), IP_3_R, and hexokinase2 (HK2) via Protein kinase B (Akt), regulating the structural and functional integrity of MAMs, calcium flux, and energy metabolism, respectively. Depletion of mTORC2 results in MAMs integrity disruption and exhibits increased gluconeogenesis, hyperinsulinemia, and impaired glucose tolerance [4]. This suggests that mTORC2 may serve as a hub receiving external signals, producing specific responses to different stimuli, thereby regulating MAMs structure and function. Additionally, decreased mTORC2 activity in mice during exercise led to reduced muscle glucose uptake, but exercise activated mTORC2 in mouse muscles and increased muscle PGC-1α expression [62,63]. This implies that exercise likely promotes MAMs structural or functional repair by activating mTORC2, thereby increasing mitochondrial biogenesis caused by upregulation of PGC-1α expression, ultimately alleviating IR.

MAMs serve as sites for mitochondrial fission, and exercise regulates the expression of mitochondrial fission-related factors [Dynamin-Related Protein 1(Drp1) and mitochondrial fission protein 1 (Fis1)] associated with MAMs structure and function. Moore et al. found that the Ser616 site of mouse Drp1 is transiently activated during acute endurance exercise [63]. Fis1 mRNA levels rapidly increase within 30 min of low-intensity continuous exercise. Studies also indicate that specific deletion of mitochondrial outer membrane protein Fis1 in skeletal muscles under exhaustive exercise leads to impaired mitochondrial function and the swelling of the sarcoplasmic reticulum (SR/ER) [64], suggesting a potential involvement of Fis1 in the regulation of MAMs by exercise. Unfortunately, current research has not definitively established an association between Fis1 and the formation of MAMs. However, studies have shown that the absence of FUN14 domain-containing 1 (FUNDC1) inhibits the expression of Fis1, resulting in excessive mitochondrial fusion. The expression of FUNDC1 is crucial for MAMs formation, and exercise has been demonstrated to upregulate FUNDC1 expression. Therefore, exercise may potentially increase MAMs structural repair by upregulating FUNDC1 expression, leading to increased Fis1 expression, inhibition of mitochondrial excessive fusion, and ultimately alleviation of insulin resistance.

Mitochondrial fusion-related molecules [Mitofusin 1 (Mfn1), Mitofusin2 (Mfn2), and Opticatrophy 1 (OPA1)] localize to MAMs and participate in regulating MAMs structure and function. Particularly, Mfn2 deficiency leads to structural disruption of MAMs, excessive mitochondrial fission, and inhibition of insulin sensitivity. Ding et al. found significant increases in Mfn1 and Mfn2 mRNA levels after 3 and 12 h, respectively, during low-intensity continuous exercise [65]. However, their levels remain unaffected during acute endurance exercise and post-exercise recovery [63]. This suggests that exercise may potentially promote the structural repair of MAMs, prevent excessive mitochondrial fission-induced IR, and achieve this by upregulating the expression of Mfn2. And, indicating potential differences in the regulatory effects of different exercise intensities and frequencies on MAMs. Research also suggests that OPA1 expression decreases in skeletal muscles of T2DM patients, while high-intensity high-volume training (HIHVT) can upregulate OPA1 expression, promoting mitochondrial fusion [66]. The possible mechanism of MAMs intervening in IR through dynamic regulation of mitochondrial fission–fusion cycles has been explained in the preceding section (Section 3.3.2). Moreover, when MAMs integrity is compromised, the balance of mitochondrial fission–fusion cycles is disrupted, suggesting that changes in MAMs structure may occur before mitochondrial dynamic alterations. In summary, it can be inferred that the enrichment of MAMs during nutritional or energy overload, as well as the structural disruption of MAMs in conditions such as IR or T2DM, can lead to the breakdown of mitochondrial fission-fusion mechanisms, resulting in IR. Exercise may promote the remodeling of the MAMs network by regulating the expression of MAMs-related molecules such as Adenosine 5′-monophosphate (AMP)-activated protein kinase (AMPK), FUNDC1, and Mfn2, thereby restoring the balance of mitochondrial fission and fusion, and ultimately alleviating IR.

During IR, mitochondrial function is impaired, accompanied by decreased mitophagy activity, while reactive oxygen species (ROS) generated during exercise can activate mitophagy to clear damaged mitochondria, enhancing mitochondrial function and alleviating IR [67]. This suggests that exercise may alleviate IR by increasing mitophagy and enhancing mitochondrial function. The structural and functional integrity of MAMs is crucial to maintaining appropriate mitophagy under insulin-resistant conditions. Therefore, exercise-mediated alleviation of IR through increased mitophagy likely involves MAMs. Studies have found that the mitophagy-related protein Parkin not only participates in regulating mitophagy but also contributes to MAMs formation. In fibroblasts with Parkin mutations, the structural and functional integrity of MAMs is disrupted. Overexpression of Parkin enhances MAMs structure and function, promoting Ca^2+^ transfer from the endoplasmic reticulum (ER) to the mitochondria and increasing Adenosinetriphosphate (ATP) production in mitochondria [68]. In previous research, we found that endurance exercise increased the adaptive expression of mitochondrial autophagy-related molecules—PTEN induced putative kinase 1(PINK1), Parkin, Nix, and Bcl2/adenovirusE1B19kDainteractingprotein3 (BNIP3)—mRNA levels in nutritionally obese mice [69]. This suggests that exercise may promote MAMs formation by upregulating Parkin expression, thereby enhancing mitophagy and alleviating IR.

Under high-fat conditions, downregulation of the FUNDC1 gene inhibits the formation of MAMs, impairs the mitophagy mechanism, and leads to more severe obesity and IR [14,41]. Although research indicates a significant increase in FUNDC1 and FUNDC1-related MAMs formation high glucose stimulation, this may partially compensate for the deficiency in mitophagy. However, at this time, the number of functionally damaged mitochondria far exceeds normal conditions. Therefore, a higher level of mitophagy is required to eliminate damaged mitochondria and restore mitochondrial function. It was shown that exercise, on the other hand, upregulated FUNDC1 expression, with the changes being more pronounced in the 4-week high-intensity exercise group compared with the 4-week moderate-intensity exercise group [70]. This suggests that exercise may also cause a compensatory increase in the formation of MAMs by upregulating the expression of FUNDC1 to obtain higher levels of mitophagy, thereby alleviating high-fat diet-induced IR. Different exercise intensities may produce varying effects.

Numerous studies have demonstrated that the dysregulation of dynamic flux in mitochondrial remodeling is associated with metabolic disorders and insulin sensitivity. As mitochondrial fission–fusion, biogenesis, and mitophagy processes are interdependent and MAMs are involved in these processes, MAMs are likely a key hub linking mitochondrial fission, fusion, biogenesis, and mitophagy. While the specific mechanisms by which exercise regulates MAMs need further investigation, strategies aimed at enhancing the capacity of the mitochondrial lifecycle through exercise-induced MAMs regulation may effectively counteract diseases related to metabolic dysfunction.

### 4.4. Mitochondria-Associated Membranes-Associated Endoplasmic Reticulum Stress Mediates Exercise Intervention in Insulin Resistance

Exercise alleviates insulin Resistance (IR) by inhibiting endoplasmic reticulum stress (ERS). Aerobic exercise may increase the activity of Chaperonin Containing TCP1, Subunit 2 (CCT2) protein through the mammalian target of rapamycin (mTOR)/Ribosomal protein S6 kinase beta-1 (S6K1) signaling pathway, enhancing protein-folding efficiency, thereby reducing unfolded protein response (UPR) and ultimately alleviating IR [71,72]. Swimming training reduced the phosphorylation levels of Protein kinase R-like endoplasmic reticulum kinase (PERK) and the alpha subunit of eukaryotic initiation factor 2 (eIF2α) in adipocytes and liver cells of high-fat diet-fed rats, thereby alleviating endoplasmic reticulum stress (ERS) [73]. Consistently with this, phosphorylation levels of Inositol-requiring enzyme-1α (IRE-1α) and c-Jun N-terminal kinase (JNK) in the liver of high-fat-fed mice decreased after 16 weeks of treadmill training. Resistance exercise also reduced the phosphorylation of JNK in cells of middle-aged and elderly men and alleviated IR [74].

The disruption of mitochondria-associated membranes (MAMs) integrity is closely related to type 2 diabetes mellitus (T2DM)-induced glucose intolerance, mitochondrial dysfunction, and intense ERS. Disruption of MAMs integrity aggravates ERS, leading to IR [45]. Swimming exercise can significantly improve glucose intolerance induced by T2DM and alleviate ERS response [58], indicating that exercise may alleviate IR by repairing the MAMs structure and inhibiting ERS. Research has found that Mfn2 expression was upregulated after two weeks of low-load high-intensity exercise training in diabetic patients [75]. Cartoni’s study also confirmed that exercise training can upregulate Mfn2 in human skeletal muscle cells [76]. Mfn2 is an important tethering protein of MAMs, and its upregulated expression promotes the formation of MAMs. It is speculated that exercise may alleviate insulin resistance by upregulating the expression of Mfn2, promoting the repair of MAMs structure to improve ER stress. However, ERS induced by skeletal muscle exercise can increase the release of muscle factors such as Fibroblast growth factor 21 (FGF21) and IL-6, ultimately reducing liver IR in non-alcoholic fatty liver disease [77,78]. This suggests that ERS has a dual role in exercise intervention in IR, and further exploration is needed to determine the specific conditions and degrees of ERS that are beneficial to the body. However, it is evidently clear that ERS caused by the disruption of the MAMs structure is not conducive to the maintenance of or enhancement in insulin sensitivity. Figure 3 shows the potential mechanism by which exercise intervenes in IR through MAMs.

## 5. Conclusions

Insulin resistance (IR) is a shared pathogenic mechanism implicated in various metabolism-related disorders. Extensive research has confirmed the crucial role of mitochondria-associated membranes (MAMs) in insulin signal transduction. However, the precise molecular mechanisms underlying MAMs-mediated regulation of IR remain poorly understood. Based on a comprehensive analysis of the existing literature, it is evident that MAMs play a pivotal role in maintaining cellular homeostasis by modulating calcium homeostasis, lipid metabolism, mitochondrial quality control, and endoplasmic reticulum stress (ERS), all of which substantially impact IR.

Under normal physiological conditions, the intracellular MAMs network maintains a stable dynamic balance. However, MAMs remodeling by energetic stimuli, including structural alterations (alterations in MAMs quantity) or functional impairments (changes in MAMs-associated tethering proteins), can disrupt calcium homeostasis, lipid impairments, and mitochondrial quality control or induce ERS, thereby promoting the development or exacerbation of IR. Exercise, by regulating MAMs formation and/or expression of associated molecules, has the potential to restore MAMs structure or function, thereby ameliorating dysregulation of cellular homeostasis and ultimately mitigating IR.

Potential mechanisms through which exercise intervenes in IR via MAMs–cellular homeostasis include those reported below.

MAMs–Ca^2+^ pathway: (1) Exercise stimulation triggers a certain mechanism leading to structural repair of MAMs, enhancing cellular MAMs-associated Ca^2+^ channels. This promotes mitochondrial Ca^2+^ uptake and inhibits cytoplasmic Ca^2+^ deposition, thereby alleviating IR. (2) Exercise directly upregulates the expression of MAMs-associated Ca^2+^ channel proteins (IP_3_Rs/RyRs/MCU), promoting mitochondrial Ca^2+^ uptake and reducing cytoplasmic Ca^2+^ deposition, thereby alleviating IR. (3) Exercise activates MAMs-related Ca^2+^ signaling pathways (G protein/IP_3_/IP_3_R), inducing Ca^2+^ signal transduction to facilitate glucose transporter 4 (GLUT4) translocation to the plasma membrane and promoting muscle glucose uptake, thereby alleviating IR.

MAMs–lipid metabolism pathway: (1) Exercise repairs the MAMs structure, reducing diacylglycerol (DAG) levels, thus alleviating IR. (2) Exercise reverses the structural damage to MAMs caused by obesity or type 2 diabetes mellitus(T2DM), inhibits triglyceride (TG) formation, thereby alleviating IR. (3) Exercise repairs the MAMs structure, alleviating ectopic deposition of ceramide, thereby alleviating IR.

MAMs–mitochondrial quality control pathway: (1) Exercise activates mammalian target of rapamycin complex 2 (mTORC2), promoting MAMs structural or fonctional repair, enhancing Peroxisome proliferator-activated receptor gamma coactivator 1-alpha (PGC-1α)-mediated mitochondrial biogenesis, and thereby alleviating IR. (2) Exercise regulates expression of MAMs-related molecules (FUNDC1, Mfn2, AMPK), promoting MAMs network remodeling and restoring mitochondrial fission–fusion balance, thereby alleviating IR. (3) Exercise upregulates Parkin or FUNDC1 expression, causing a compensatory increase in MAMs formation, facilitating higher levels of mitophagy, and alleviating high-fat diet-induced IR. Different exercise intensities may produce different effects.

MAMs–ERS pathway: Exercise repairs the structure of MAMs by upregulating Mfn2 expression, thereby alleviating ERS and ultimately alleviating IR.

However, there is relatively little direct evidence that MAMs mediate exercise-regulated IR. Mechanisms governing exercise-mediated modulation of the MAMs network balance require further investigation, with many questions remaining to be addressed. For example, areas requiring further clarification are quantifying the relationship between MAMs and cellular life activities; understanding how exercise promotes or inhibits MAMs formation based on the state of stimulation experienced by the body; and elucidating the impact of exercise load, intensity, and duration on MAMs regulation.

As research progresses and advanced experimental techniques develop, the true nature of MAMs hidden behind the veil is gradually being unveiled. The intricate mechanism of MAMs-mediated exercise intervention in IR will continue to be refined, offering additional possibilities for research on and treatment of metabolic diseases.

## Figures and Tables

**Figure 1 ijms-25-03196-f001:**
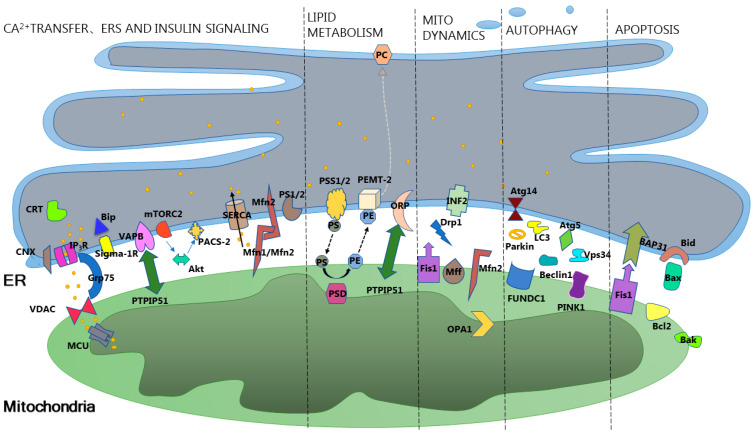
Schematic diagram of mitochondria-associated membranes structure. Ca^2+^ transfer: IP_3_R (inositol 1,4,5-triphosphate receptor); Grp75 (glucose-regulated protein 75); VDAC (voltage-dependent anion channel); MCU (mitochondrial calcium uniporter); CNX (calnexin); CRT (calreticulin); VAPB (VAMP-Associated Protein B And C); PTPIP51 (Protein tyrosine phosphatase-interacting protein 51); Bip (immunoglobulin heavy chain binding protein in pre-B cells); Sigma-1R (sigma-1 receptor); SERCA (sarco/endoplasmic reticulum Ca^2+^ ATPase); Mfn2 (Mitofusin2); Mfn1 (Mitofusin1); PS1/2 (Presenilin (PS) 1 and 2). Endoplasmic reticulum stress: Bip; Sigma-1R; CNX; CRT. Insulin signaling: mTORC2 (malian target of rapamycin complex 2); Akt (Protein kinase B); PACS-2(Phosphofurin acidic cluster sorting protein 2). Lipid metabolism: PS(Phosphatidylserine); PE (phospha-tidylethanolamine); PC (phosphatidylcholine); PSS1/2 (PS synthase-1/2); PSD (PS decarboxylases); ORP (oxysterol-binding protein-related protein); PTPIP51; PEMT-2 (phosphatidylethanolame-*N*-methyltransferase 2). Mitochondrial dynamics: Fis1 (mitochondrial fission protein 1); Drp1 (dynamin-related protein 1); Mff (mitochondrial fission factor); INF2 (ER-localized inverted formin 2); OPA1 (opticatrophy); Mfn2. Autophagy: Atg14 (autophagy-related 14); Atg5 (autophagy-related 5 homolog); LC3(Microtubule-associated protein light chain 3); Beclin1; PINK1(PTEN induced putative kinase 1); Parkin; FUNDC1 (FUN14 domain-containing 1); Vps34 (Phosphatidylinositol 3 kinase type 3 catalytic subunit). Apoptosis: BAP31 (B-Cell Receptor-Associated Protein 31); Fis1; Bax (BCL2-Associated X); Bid (BH3-Interacting Domain Death Agonist); Bcl2 (B-cell lymphoma-2); Bak (Bcl-2 homologous antagonist/killer).

**Figure 2 ijms-25-03196-f002:**
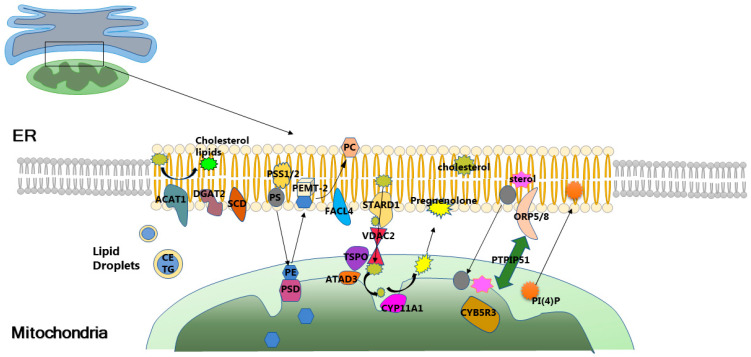
Lipid metabolism-related proteins and pathways involving mitochondria-associated membranes. Abbreviations: CE: cholesterol ester; TG: cholesterol ester; ACAT1: Acetyl-CoA Acetyltransferase 1; DGAT2: Acyl CoA: Diacylglycerol Acyltransferase 2; SCD: Stearoyl-CoA desaturase; PSS1/2: PS synthase-1/2; PS: Phosphatidylserine; PE: phospha-tidylethanolamine; PSD: PS decarboxylases; PEMT-2: phosphatidylethanolame-*N*-methyltransferase 2; PC: phosphatidylcholine; FACL4: Acyl CoA synthetase long chain family member 4; STARD1: START domain-containing protein 1; VDAC2: voltage-dependent anion channel 2; TSPO: translocator protein; ATAD3: AAA domain-containing protein 3; CYP11A1: Cytochrome P450 Family 11 Subfamily A Member 1; ORP5/8: oxysterol-binding protein-related proteins5/8; PTPIP51: Protein tyrosine phosphatase-interacting protein 51; CYB5R3: Cytochrome B5 Reductase 3; PI(4)P: Phosphatidylinositol4-phosphate.

**Figure 3 ijms-25-03196-f003:**
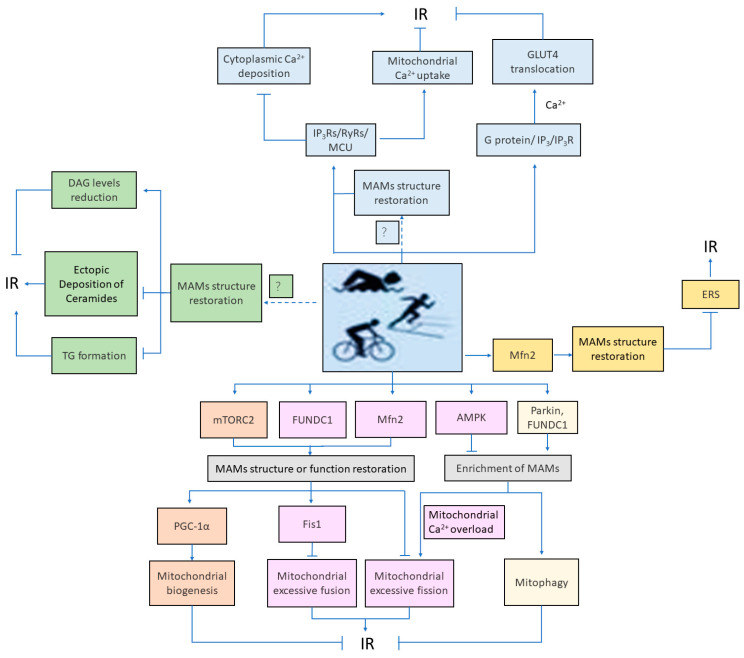
The potential mechanisms by which exercise intervenes in insulin resistance through mitochondria-associated membranes. Mechanisms of exercise intervention on insulin resistance(IR) via mitochondria-associated membranes (MAMs)-Ca^2+^ pathway: (1) Exercise stimulation triggers a mechanism that repairs MAMs structure, leading to an increase in MAMs-associated Ca^2+^ channels within cells, promoting mitochondrial Ca^2+^ absorption, and inhibiting cytoplasmic Ca^2+^ deposition, thereby alleviating IR. (2) Exercise directly upregulates the expression of MAMs-associated Ca^2+^ channel proteins (IP3Rs/RyRs/MCU), promoting mitochondrial Ca^2+^ absorption, decreasing cytoplasmic Ca^2+^ deposition, and alleviating IR. (3) Exercise activates the MAMs-related Ca^2+^ signaling pathway (G protein/IP3/IP3R), induces Ca^2+^ signal transduction, promotes GLUT4 translocation to the plasma membrane, and enhances muscle glucose uptake, thus alleviating IR. Mechanisms of exercise intervention on IR via MAMs-lipid homeostasis pathway: (1) Exercise reduces DAG levels by repairing the MAMs structure, thereby alleviating IR. (2) Exercise alleviates IR by promoting the restoration of the MAMs structure in obesity or T2DM, inhibiting TG formation. (3) Exercise alleviates IR by repairing the MAMs structure, alleviating ectopic deposition of ceramides in obese and T2DM.Mechanisms of exercise intervention on IR via MAMs-mitochondrial biogenesis pathway: Exercise promotes MAMs structural or functional repair by activating mTORC2, thereby increasing mitochondrial biogenesis caused by upregulation of PGC-1α expression, ultimately alleviating IR. Mechanisms of exercise intervention on IR via MAMs-mitochondrial dynamics pathway: (1) Exercise promotes MAMs structural repair by upregulating FUNDC1 expression, leading to increased Fis1 expression, inhibition of mitochondrial excessive fusion, and ultimately alleviation of IR. (2) Exercise promotes MAMs structural repair by upregulating Mfn2 expression, preventing excessive mitochondrial fission-induced IR. (3) Exercise alleviates high-sugar- and high-fat-induced IR by activating AMPK, downregulating FUNDC1 expression, inhibiting MAMs enrichment, preventing excessive mitochondrial fission. Mechanisms of exercise intervention on IR via MAMs-mitophagy path-way: (1) Exercise causes a compensatory increase in the formation of MAMs by upregulating the expression of FUNDC1 to obtain higher levels of mitophagy, thereby alleviating high-fat diet-induced IR. (2) Exercise promotes MAMs formation by upregulating Parkin expression, thereby enhancing mitophagy and alleviating IR. Mechanisms of exercise intervention on IR via MAMs-endoplasmic reticulum Stress pathway: Exercise alleviates IR by upregulating the expression of Mfn2, promoting the repair of MAMs structure to improve ERS.

## Data Availability

No new data were created or analyzed in this study. Data sharing is not applicable to this article.

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
