# Peer review of "Mitochondria-Associated Membranes as Key Regulators in Cellular Homeostasis and the Potential Impact of Exercise on Insulin Resistance"

_ijms, 2024, doi:10.3390/ijms25063196_

Round 1
Reviewer 1 Report
Comments and Suggestions for Authors
Li et al. provide a detailed summary of the relationship between exercise and MAMs, giving a strong rationale for further research into this area to address metabolic diseases, such as T2DM. In general, the review is clearly written, and the authors provide a lot of interesting details regarding how MAMs play a key role in the physiology of muscle and how disruption of MAMs underlies insulin resistance.
The authors highlight the deleterious effects of high fat/sugar diets on muscle physiology and correctly draw attention to dysregulation of lipid metabolism in pathological muscle. It would be informative to contrast this with the seemingly paradoxical role of perilipin 5 in the regulation of lipid metabolism in the skeletal muscle of endurance athletes (Laurens et al., 2016).
Line 216-227: The language here sounds like there is a speculation that there is a connection between MAMs and mitochondrial dynamics, particularly mitochondrial division. For over a decade, it has been established that the ER plays a key role in mitochondrial fission (Friedman et al., 2011) as well as fusion (Abrisch et al., 2020). This paragraph, and others, should be modified to reflect the current understanding in the field with regard to inter-organelle communication and mitochondrial dynamics. It is important to note that, in addition to the ER, other organelles appear to associate with the MAMs in the context of mitochondrial dynamics – for example, lysosomes (Wong et al., 2018).
Lines 246-249: The phrase “higher equilibrium state” is ambiguous and is not standard in the context of membrane remodelling at the MAMs. The authors are attempting to express that nutrient excess disrupts normal MAM architecture, and this leads to altered mitochondrial dynamics. However, the exact reasons that mitochondria fragment as a result of persistent nutrient overload is still not clear. Some literature suggests that nutrient overload is associated with mitochondrial depolarization: e.g., depolarization leads to cleavage of OPA1 and impaired fusion, together with recruitment of DRP1 to promote mitochondrial division, but the exact reasons that mitochondrial morphology changes in this way is still unclear.
Lines 251-260: The authors should cite the primary literature, here – e.g., Twig et al., 2008. The targeting of depolarized mitochondria to the lysosome, as part of the broader mitochondrial life cycle, is relevant to the discussion of mitophagy in the context of MAMs and exercise.
In general, this review would be improved if the authors make additional efforts to cite the primary literature.
Bibliography:
Friedman JR, Lackner LL, West M, DiBenedetto JR, Nunnari J, Voeltz GK. ER tubules mark sites of mitochondrial division. Science. 2011 Oct 21;334(6054):358-62. doi: 10.1126/science.1207385. Epub 2011 Sep 1. PMID: 21885730; PMCID: PMC3366560.
Abrisch RG, Gumbin SC, Wisniewski BT, Lackner LL, Voeltz GK. Fission and fusion machineries converge at ER contact sites to regulate mitochondrial morphology. J Cell Biol. 2020 Apr 6;219(4):e201911122. doi: 10.1083/jcb.201911122. PMID: 32328629; PMCID: PMC7147108.
Wong YC, Ysselstein D, Krainc D. Mitochondria-lysosome contacts regulate mitochondrial fission via RAB7 GTP hydrolysis. Nature. 2018 Feb 15;554(7692):382-386. doi: 10.1038/nature25486. Epub 2018 Jan 24. PMID: 29364868; PMCID: PMC6209448.
Laurens C, Bourlier V, Mairal A, Louche K, Badin PM, Mouisel E, Montagner A, Marette A, Tremblay A, Weisnagel JS, Guillou H, Langin D, Joanisse DR, Moro C. Perilipin 5 fine-tunes lipid oxidation to metabolic demand and protects against lipotoxicity in skeletal muscle. Sci Rep. 2016 Dec 6;6:38310. doi: 10.1038/srep38310. PMID: 27922115; PMCID: PMC5138838.
Comments on the Quality of English Language
The English is largely fine. There are some typos, and the style could be improved here and there, but overall the text is pretty good.
Reviewer 2 Report
Comments and Suggestions for Authors
This review article discusses the importance of mitochondria-associated membranes (MAMs) in the communication between mitochondria and the endoplasmic reticulum (ER). It highlights the role of MAMs in insulin signal transduction and their regulation of calcium and lipid homeostasis, mitochondrial quality control, and endoplasmic reticulum stress. The article evaluates recent research findings suggesting that exercise can remodel the structure and function of MAMs by modulating the expression of associated molecules. This remodeling may restore cellular homeostasis and improve insulin resistance. The insights provided offer potential implications for the study and treatment of metabolic disorders associated with insulin resistance.
Major points.
Despite the review covering a broad range of the connection between MAMs and insulin resistance, more information could be added. The authors barely mention the role of MAMs in the regulation of inflammation, despite there being a considerable number of studies dedicated to this topic.
Minor points:
Lines 29 and 33. IR (insulin resistance) and ERS (endoplasmic reticulum stress) should be spelled out upon their first mention in the introduction, not in the abstract only.
Line 114 and onwards. The authors did not use superscript and subscript characters where necessary, such as Ca2+ and so on.
